# Identification of Patulin from *Penicillium coprobium* as a Toxin for Enteric Neurons

**DOI:** 10.3390/molecules24152776

**Published:** 2019-07-30

**Authors:** Benjamin Brand, Nicolai M. Stoye, Malena dos Santos Guilherme, Vu Thu Thuy Nguyen, Julia C. Baumgaertner, Anja Schüffler, Eckhard Thines, Kristina Endres

**Affiliations:** 1Department of Psychiatry and Psychotherapy, University Medical Center Johannes Gutenberg-University Mainz, 55131 Mainz, Germany; 2Institut für Biotechnologie und Wirkstoff-Forschung gGmbH, 67663 Kaiserslautern, Germany; 3Institute for Microbiology and Wine Research, Johannes Gutenberg University Mainz, 55128 Mainz, Germany

**Keywords:** enteric nervous system, fungi, fusarium, gastrointestinal system, microbiome, mycotoxins, *Penicillium*

## Abstract

The identification and characterization of fungal commensals of the human gut (the mycobiota) is ongoing, and the effects of their various secondary metabolites on the health and disease of the host is a matter of current research. While the neurons of the central nervous system might be affected indirectly by compounds from gut microorganisms, the largest peripheral neuronal network (the enteric nervous system) is located within the gut and is exposed directly to such metabolites. We analyzed 320 fungal extracts and their effect on the viability of a human neuronal cell line (SH-SY5Y), as well as their effects on the viability and functionality of the most effective compound on primary enteric neurons of murine origin. An extract from *P. coprobium* was identified to decrease viability with an EC_50_ of 0.23 ng/µL in SH-SY5Y cells and an EC_50_ of 1 ng/µL in enteric neurons. Further spectral analysis revealed that the effective compound was patulin, and that this polyketide lactone is not only capable of evoking ROS production in SH-SY5Y cells, but also diverse functional disabilities in primary enteric neurons such as altered calcium signaling. As patulin can be found as a common contaminant on fruit and vegetables and causes intestinal injury, deciphering its specific impact on enteric neurons might help in the elaboration of preventive strategies.

## 1. Introduction

Latest research indicates that the gastrointestinal tract might play an even bigger role in health and disease of mammalian organisms than assumed in the past. Especially the gut is regarded to be of great importance due to its unique attributes: it harbors a vast number of microorganisms, plays a crucial role in digestive processes, functions as a barrier and thereby as a part of the immune system [1], and lastly, comprises its own nervous system. Already a century ago, it was discovered that intestinal functions are controlled almost autonomously from the central nervous system (CNS) by a local nerve plexus [2]. This so-called enteric nervous system (ENS) is also referred to as “the brain in the gut” [3], and consists of an estimated 200–500 million neurons [4] with 14 functionally- and morphologically-different neuron types found, for instance, in the ENS of the guinea-pig [5]. Although the ENS can function independently, the CNS and ENS show closely-linked bidirectional communication in the so-called “gut-brain-axis”, which includes the vagus nerve, but also endocrine and immune links [6]. Due to the connection of cognitive and emotional areas of the brain with gut functions, the “gut-brain-axis” is seen as a therapeutic target for gastrointestinal but also psychiatric disorders [6].

In particular, the influence that the microorganisms inhabiting the gut exert locally on the ENS, but also on the entire system, is of great interest regarding more systemic diseases or functions. A research initiative by the United States National Institute of Heath called “Human Microbiome Project” aimed at identifying and characterizing the human microbiome describes the variety of microbial communities inhabiting the human body and their respective genes [7]. The majority of the mapped sequences within this project originated from bacteria, i.e., over 99%, followed by eukaryotes and archaea [7].

As bacteria are found to be the predominant domain of the human microbiome, there have been fewer studies regarding the much smaller domain of fungi in the so-called mycobiome. Though fungal genes only make up a share of 0.3–1% of the human microbiome [8,9], fungi still constitute a bigger biomass than suggested by the small number of available genomes, as a typical fungal cell is more than 100 times bigger than a typical bacteria cell [10]. Therefore, the influence exerted by fungi on the host organism and the homeostasis in the gut should not be underestimated. The composition of the gut mycobiome has not yet been definitively determined, but several studies have identified mainly Saccharomyces and Candida belonging to the group of yeasts, as well as the genus *Aspergillus* and *Penicillium*, which are characterized as molds [9,11,12]. Much like commensal bacteria, fungi seem to have a promoting effect on the hosts’ health. Colonizing the gut of antibiotic-treated mice with either *Saccharomyces cerevisiae* or *Candida albicans* was observed to recapitulate the protective benefits of intestinal bacteria after induction of a severe colitis or virus infection of the host [13], and the often used probiotic *Saccharomyces boulardii* showed anti-inflammatory effects in patients with chronic inflammatory bowel disease [14]. Still, fungi are a well-known cause for various diseases, ranging from simple nail and skin infections to severe lung infections by e.g., *Aspergillus*. The latest findings indicate that the gut mycobiome may also be involved in intestinal diseases such as irritable bowel syndrome and chronic inflammatory bowel disease [15,16,17,18].

As the impact of fungal metabolites on the ENS is mostly unknown, we analyzed a small library of extracts from dung-associated fungi to identify potential new substances interfering with the physiology of the gut. We were able to identify an extract from *Penicillium coprobium*, which showed severe neurotoxic effects even at very low dosages in SH-SY5Y cells, and to the same extent in primary enteric neurons of murine origin. Further analysis characterized the toxic compound in the extract to be patulin, and we were able to recapitulate the observed effects on cell viability and functional pathways with purified patulin. Additionally, we could show reduced neurite outgrowth of primary enteric neurons evoked by a low dosage of patulin. To our knowledge, this is the first study showing the effects of patulin on primary enteric neurons, thus providing new insight into the effects of this mycotoxin on gut physiology. These findings might therefore contribute to preventive strategies against mycotoxicosis by patulin.

## 2. Results

We aimed at identifying compounds with neurotoxic effects from a bank of 320 fungal extracts, including organisms found on dung (gut-prone or saprophytic).

### 2.1. Toxic Effect of Fungal Extracts on Neuronal Cells

To analyze the potential toxic impact of fungal components on the viability of neuronal cells, we chose SH-SY5Y cells that serve as representatives of neurons in many investigations [19]. Cells were treated for 24 h with the extracts in an initial concentration of 5 ng/µL and ATP content, mirroring viability, measured with the Cell Titer Glo Assay (Promega, [20,21]). A hit was defined if within at least three independent experiments, treatment with the extract resulted in a value <100%-3xSD% and a SD between experiments of <10%. The mean of solvent (DMSO)-treated cells was therefore set to 100% and a three-fold SD (deviation for all substances tested) was used to obtain reasonable robust hits [22]. Two extracts were identified as being neurotoxic: one from *Fusarium sp.* and one from *Penicillium coprobium* (Figure 1).

Cells were incubated with the extracts at a concentration of 5 ng/µL for 24 h. Viability was assessed by measuring ATP content with a luminescent assay in at least three independent experiments. Results with too high a standard deviation and potential hits were tested in two additional tests. Shown are means ±SD in % of the cells treated with DMSO (1%; solvent). Hits were identified by choosing effect sizes < mean of controls + 3xSD, and are indicated by red dots.

Subsequently, we conducted a dose-response analysis and achieved for both extracts a dependency of observed reduction in viability and administered dosage (Figure 2a,b). To avoid higher doses of the solvent DMSO, which itself can exert a toxic effect on cells (e.g., [23]), the concentration of the *Fusarium*-derived extract could not be increased to more than 10 ng/µL (Figure 2a). For the extract derived from *P. coprobium*, we could demonstrate an EC_50_ of 0.23 ng/µL for SH-SY5Y cells (Figure 2b). Mycotoxins can act on the host’s physiology in many ways, such as building DNA-adducts as aflatoxins or inhibiting protein synthesis as ochratoxins (summarized in [24]). Other mycotoxins evoke production of ROS and thereby harm host cells (e.g., [25]). When measuring ROS production, the initial concentration of both extracts evoked significantly elevated ROS production (Figure 2c)—comparable or even higher than the positive control rotenone [26]. Because of 5 ng/µL of *P. coprobium* extract being immensely toxic to the cells, we also included lower concentrations of 0.1 to 1 ng/µL and found an increase of ROS production with 1 ng/µL, while lower concentrations had no effect. As the effect exerted by *Fusarium sp.* was comparably low, and the *P. coprobium* extract seemed to be a more promising candidate for impacting neuronal cells, we focused only on the latter in the following experiments.

### 2.2. Impact of P. coprobium Extract on Primary Enteric Neurons

SH-SY5Y cells have been widely used for investigations regarding neurotoxicity, and can even be differentiated into a functionally-mature neuronal phenotype. However, these cells are still of cancer origin. Therefore, we aimed at recapitulating our experiments in a more physiologically-relevant model. Enteric neurons are the neurons that first come into contact with the mycotoxins when being ingested with the diet. When enteric neurons of murine origin were incubated with 5 ng/µL of *P. coprobium* extract, they showed statistically significant reduction in viability that also became obvious when looking at them through the microscope (5% of solvent-treated cells, *p* < 0.0001; data not shown; Figure 3a). 0.23 ng/µL, the EC50 assessed for SH-SY5Y cells, resulted in no reduction of viability (92% of solvent-treated cells, *p* < 0.33, data not shown).

However, cells displayed an altered morphology, and the interconnecting axon bundles between the ganglia normally characterizing these cells in culture occurred much than under control conditions (Figure 3a). The intestinal tract serves not only to harvest nutrients, but also to protect against harmful toxins and pathogens. Therefore, a more robust phenotype of these enteric neurons as compared to the secondary SH-SY5Y cell line might be assumed. Additionally, we cannot exclude the hypothesis that the primary culture also contains glial cells to a certain extent. For example, glial-derived neurotrophic factors could act to protect neuronal cells from apoptosis during inflammatory states [27]. Nevertheless, the altered morphological phenotype was accompanied also by functional deficit: neurons incubated with a subtoxic (0.23 ng/µL) concentration of the extract revealed increased calcium signals due to depolarization, as compared to control-treated cells (Figure 3b). The delicate dynamic balance between extracellular and cytosolic calcium does not only guarantee its function as a fundamental second messenger and trigger for neurotransmitter release; calcium also controls synaptic plasticity, and even gene expression in neurons. Disruptions in the calcium regulatory mechanism are involved in the development of irreversible neuronal damage, as, for example, described for Alzheimer’s disease (e.g., [28]).

A dose-response curve conducted in the primary cell line finally resulted in a four-fold increase in EC50 compared to the SH-SY5Y cell line (1 ng/µL vs. 0.23 ng/µL, Figure 3c).

### 2.3. Identification of Toxic Compounds Derived from P. coprobium Extract

To identify the neurotoxic fractions within the *P. coprobium*-derived extract, dried material from two independent fractionations was dissolved in DMSO, diluted with DMSO according to the initial experiments, and tested on SH-SY5Y cells. Three fractions (B2-4) showed significant impact on viability of the cells after 24 h of incubation (Figure 4a) and were congruent with a broad peak observed at 210 nm (4.5 min) of the fractionating HPLC run (chromatogram after fractionation: Figure 4b). The spectrum of the selected peak was compared to library data and found to be identical with that of patulin, which was available as a standard, in terms of retention time, mass, and UV spectra (Figure 4c). The mycotoxin patulin (4-hydroxy-4*H*-furo [3,2c]pyran-2[6*H*]-one) is a secondary metabolic product of species from genera such as *Penicillium*, *Aspergillus*, and *Byssochlamys* (reviewed in [29]). A recent review [30] summarizes the systemic effects of patulin after dietary exposure in mammals to include intestinal injury, intestinal ulcers, inflammation, bleeding, and a decrease in transepithelial resistance, rise in ALT, AST, and MDA after reaching the liver, degeneration of glomeruli and renal tubules in the kidney, and neurotoxicosis and neuronal degeneration within brain. 

Subsequently, we tested purified patulin for its ability to act toxic on SH-SY5Y cells; as shown in Figure 5a, 22 µM patulin (equals the 5 ng/µL concentration of the extract of *P. coprobium*) only achieved 50% of the toxic effect, while the corresponding concentrations flanking the EC50 of the extract (2.2 and 0.44 µM) had no significant effect. To further analyze this deviation of pure patulin with regard to the patulin-containing extract, we measured ROS production in cells treated with the two highest patulin concentrations: 22 and 4.4 µM (Figure 5b). 22 µM patulin resulted in 150% of ROS release in comparison to solvent-treated cells within the last 6 h of incubation, while 4.4 µM had no effect. This confirms that patulin alone has a weaker impact on the cells than the extract from *P. coprobium* (see Figure 2c).

### 2.4. Lack of Toxicity of P. coprobium Extract on Fecal Bacteria

Gut function is controlled by both, the host’s intestinal cells and the microbial commensals and their interplay (e.g., [31]). In this regard, a mycotoxin might act not only on enteric neuronal cells, but also on the gastrointestinal microbial partners. To analyze this, we acutely treated fecal microbiota from mouse with *P. coprobium* extract at 5 ng/µL and measured ATP content as a proxy for cell viability (Figure 6a, left graph). No differences due to treatment with the fungal extract could be observed. Moreover, we counted colony forming units on Lactobacteriaceae- and Enterobacteriaceae-selective cultivation plates. For both, no statistically-significant difference in growth as compared to solvent-treated samples was obtained, even though a slight tendency for growth reduction might be assumed in the Enterobacteriaceae group (*p* = 0.26; Figure 6a).

With the knowledge of patulin being the toxic component, one would have expected an effect on the gut microbiota. Patulin has been shown also to have antimicrobial activity, which led to a trial which used it against common cold [32]. However, its severe side effects resulted in cancelling those attempts. Using bacteria from mouse feces, 22 µM of purified patulin (comparable to 5 ng/µL of extract) were not sufficient to exert a toxic effect (Figure 6b), and also did not significantly reduce colony forming units of Enterobacteriaceae (*p* = 0.15). On the contrary, a 20% reduction of Lactobacteriaceae was observed.

### 2.5. Impact of Patulin on Physiological Functions of Enteric Neurons

There are only few reports analyzing the effect of patulin in vitro on intestinal cells, all of which used secondary cell lines such as HT29 and Caco-2 [33,34]. To our knowledge, the effects exerted on primary enteric neurons have not been investigated in detail so far. We therefore subsequently investigated the impact of purified patulin on central pathways of enteric primary neurons such as general viability, excitability, and glucose consumption (Figure 7). While the extract nearly abolished all ATP from the cells, patulin achieved a more than 50% reduction of ATP at 10 µM and led to a statistical reduction also at 0.5 µM (Figure 7a). The impaired calcium signaling obtained for the extract (Figure 3b) could also be recapitulated by purified patulin, even if 0.5 µM only resulted in a trend (Figure 7b). This increased calcium entry has also been found for human erythrocytes after incubation with patulin [35]. Decades ago, patulin was shown to affect glucose uptake into cells [36]. As enteric neurons together with enteroendocrine cells represent the most important cells for sensing intraluminal contents such as glucose [37,38], an impairment of this function might have dramatic consequences for nutritional resorption. Ten of patulin did not alter glucose uptake in a statistically-significantly manner, but we have to take into account the already severe impact on metabolism by this high dosage as demonstrated by ATP reduction. With a low dosage of 0.5 µM, patulin reduced the cellular glucose content to 83% of control-treated cells (Figure 7c). As we already gained the impression that subtoxic concentrations of the patulin-containing extract affected neuronal morphology (Figure 3a), we next assessed quantitatively the amount of neurite outgrowth by a cellular transwell culturing system: total neurite mass was reduced to about 70% of control-treated cells after 24 h of incubation with 0.5 µM patulin (Figure 7d).

## 3. Discussion

The influence of intestinal fungi on the host may be caused directly by their metabolites. Additionally, bacteria-fungi interactions are hypothesized to affect the host indirectly [39], as fungi are known to produce antibiotics, with penicillin as a prominent example, and bacterial short-chain fatty acids such as butyrate can inhibit the germination of *C. albicans* [40]. Thirdly, there might be direct interactions with the cells of the gut, as fungi such as *Saccharomyces* and *Candida*, but also *Penicillium*, are able to synthesize short-chain fatty acids [41]. Many fungi are also known to produce various mycotoxins such as aflatoxin, ochratoxin A, and zearalenone which are severely harmful to the hosts’ health [42]. Acute or chronic mycotoxicosis is observed to cause impairments ranging from cancerogenic effects to death [42,43]. Here, we identified an extract from the mycelium of *Penicillium coprobium* (for a phenotypic characterization see [44]) as a new toxic agent, acting on enteric neurons of the mouse. That *P. coprobium* belongs to the normal commensal community of mammalian gut has to be doubted, as, to our knowledge, it has only been identified in fecal samples so far [45], and not in chyme samples. 

Coprophilic fungi including Penicillia share a competitive substrate with many other microorganisms, and may therefore benefit from producing bioactive compounds such as patulin [46]. A growth inhibitory property towards other microorganisms such as *E. coli* or *B. subtilis* was reported decades ago (reviewed in [47]). However, an extract from *P. coprobium* as well as patulin displayed no major impact on the fecal microbial community. Only for purified patulin a significant reduction of colony-growth of 20% occurred for Lactobacteriaceae. The discrepancy between the lack of effect of the extract and partial toxicity of its purified compound could potentially result from other extract components. For example, sulfhydryl groups have been identified as potent destabilizers of patulin [48]. Recently, *Lactobacillus acidophilus* and *Lactobacillus plantarum* have been shown to efficiently remove patulin from apple juice samples [33]; therefore, it is plausible that bacteria of the fecal community would also be capable of detoxification to a certain extent, and that potentially, Lactobacteriaceae are sensitive to patulin. Additionally, patulin has been shown to decompose at pHs above 6 [49]. As feces of mice typically show a pH of 7.4 (e.g., [50]), we cannot exclude that patulin was already destabilized and could not exert a highly inhibiting function in this physiological condition. The local influence of intact patulin molecules on single microbial niches in the gut cannot be fully excluded, as only representative genera derived from feces were investigated in this study. However, we concluded that the gastrointestinal effects of the extract and patulin dominantly may consist of their impact on gut cells themselves.

Patulin, which made up 68% of the extract of *P. coprobium*, contributed substantially to toxicity and ROS-production in the neuronal model cell line SH-SY5Y. We were not able to identify further toxic compounds by fractionation; however, a reaction with patulin with one of the compounds from the extract can also be assumed. Patulin has been shown to react with e.g., glutathione and its metabolite *N*-acetyl-l-cysteine, due to its electrophilic properties via non-enzymatic reactions [51]. This might lead to an altered stability of patulin or better penetrance into the mammalian cell.

The influence of patulin on barrier function of the gut has been investigated using intestinal epithelial cell model Caco-2: trans-epithelial electrical resistance (TER) was reduced by 50 µM patulin and the tight junction protein ZO-1 was strongly reduced [52]. Experiments performed using rat colonic mucosae showed that a concentration as high as 500 μM was needed to sustainably decrease TER [34]. However, here we were able to demonstrate that even very low concentrations of patulin can affect important functions of enteric neurons. The WHO-recommended maximum level of patulin is 50 ng/mL in apple-derived beverages [53], which is just slightly below the low dose used in our investigation (0.5 µM = 77 ng/mL). As the enteric nervous system autonomously regulates many gastrointestinal functions such as peristalsis and resorption, such impairment could explain the tremendous effects of the mycotoxin such as distension, ulceration, and hemorrhage when ingested via the oral route. 

Due to the observed neurotoxic effects, we also propose that the usage of patulin as a chemopreventive for lung cancer through inhibition of the NF-κB pathway as suggested by a recent publication [54] should be viewed with caution. Though it was discovered that 1.5 µM patulin can inhibit roughly 90% of NF-κB activity in HEK cells, and therefore could serve as a therapeutic agent [54], our findings show that even a third of the tested concentration (0.5 µM) can significantly reduce the viability of neuronal cells. As sensory nerves innervating the lung play an important role in regulating cardiopulmonary functions and defense reflexes [55], severe side effects on neuronal cells using patulin with a therapeutic approach cannot be ruled out.

## 4. Materials and Methods 

### 4.1. Fungal Extracts

Fungal extracts were obtained by cultivating individual fungi in 0.5 L YMG medium (4 g/L yeast extract, 10 g/L malt extract, 10 g/L glucose, adjusted to pH 5.5 before autoclaving) in 1 L flasks with one baffle. Fermentations were terminated when the glucose was used up and culture broth was separated into mycelium and culture fluid by filtration. Extracts were generated from the mycelium after freeze-drying by extraction with acetone/MeOH and from the culture filtrate by direct extraction with ethyl acetate. After evaporation of the solvents, the extracts were dissolved in MeOH at a known concentration and stored at −20 °C. Extracts were received as dried material (400 µg per well) and diluted in DMSO (AppliChem GmbH, Darmstadt, DE) to 0.5 µg/µL light-protected and with 10 min of shaking at room temperature and 500 rpm. Diluted extracts were plated on clear 96-well plates at 2 µL per well and stored light-protected at −20 °C until further use.

### 4.2. Cell Culture

Cells were maintained in humidified air (95%), 5% CO_2_ and 37 °C. Cultivation of SH-SY5Y human neuroblastoma cells (ATCC: CRL-2266) was performed using DMEM/F12 (Gibco by Life technologies Carlsbad (CA), USA) containing 10% FCS (Gibco by Life technologies, Carlsbad (CA), USA) and 1% l-glutamine (Sigma Aldrich, St. Louis (MO), USA) as well as 1% penicillin/streptomycin (Sigma Aldrich, St. Louis (MO), USA). Cells were passaged twice a week 1:2–1:4. 

Enteric neurons were prepared from 2–3-month-old C57B6/J OlaHsd mice (Envigo) following a previously published protocol [56]. In brief, the abdominal skin and cavity was opened with scissors to reveal the internal digestive organs. The gut was removed and the small intestine and the colon were flushed with ice-cold Krebs solution. The attached mesenterium was removed to open the longitudinal muscle. The longitudinal muscle was teased away from the circular muscle with a cotton swab using horizontal strokes. Tissue containing the myenteric plexus was rinsed and digested for 60 min at 37 °C and 300 rpm with Collagenase Type 2 (1.3 mg/mL, Worthington Biochemical Corporation, Lakewood, NY (USA)). Cells were collected by centrifugation and filtered through a 100 μm cell strainer (Greiner bio-one). Cells were then diluted in culture medium (Neurobasal A, supplemented with GDNF (Origene), B-27, FBS, l-Glutamine, and Penicillin-Streptomycin (all Life Technologies)) and incubated for four days at 37 °C, 5% CO_2_, 95% air humidity. Cells were monitored by light-microscopy using the Evos Digital microscope (ThermoFisherScientific). Donor animals were balanced for sex (50% male) and at least 4 donor animals were used for each experiment in technical replicates. All experiments were conducted in accordance with the official regulations for the care and use of laboratory animals, and approved by local authorities (University of Mainz, Germany). 

### 4.3. Cytotoxicity Test

Cytotoxic effects of the fungal extracts were assessed by using the CellTiter-Glo Assay (Promega, Mannheim, Germany) in 96-well formats (white, flat glass bottom, Greiner). In experiments with SH-SY5Y cells, drug concentration was set as indicated in 50 µL culture medium (without 1% penicillin/streptomycin) and 40,000 cells per well were used. Experiments with enteric neurons were conducted using the respective drug concentration in 100 µL culture medium. In both cases, an incubation period of 24 h was tested. Relative luminescence units were recorded using the Fluostar Omega plate reader (BMG). Dose-response curves were conducted as described with a DMSO volume of 1% in the controls and in the extract-treated wells.

### 4.4. ROS Test

To assess production of reactive oxygen species (ROS) evoked by fungal extracts, cells were treated for 18 h as described, then H_2_O_2-_ and detection substrate were added as indicated by the vendor (Promega, Mannheim, Germany) for the final 6 h of incubation. Relative luminescence units were recorded as described for the toxicity assay.

### 4.5. Calcium Signaling

To investigate, whether the selected fungal extract exerted an effect on intracellular calcium signaling in enteric neurons, a calcium flux assay was performed using the fluorescent calcium indicator Calbryte 520 AM (AAT Bioquest, Sunnyvale (CA), USA). Cells were treated with drug concentrations as indicated in 100 µL culture medium and incubated for 24 h at 37 °C, 5% CO_2_ and humidified air (95%). An equal volume of a 10 µM dye working solution (in HHBS) was added to the cells and the dye-loading plate was incubated in the cell incubator for 45 min, and subsequently for 15 min at room temperature, as indicated by the vendor. Relative fluorescent units were measured using the Fluostar Omega plate reader (BMG) with an orbital summation. A base level of fluorescence was recorded; then, the cells were depolarized with 50 mM KCl solution and the difference in relative units was calculated. 

### 4.6. Measurement of Glucose Content

Glucose uptake in enteric neurons was assessed using the Glucose-Glo Assay (Promega, Mannheim, Germany) in 96-well formats (white, flat glass bottom, Greiner). Cell were treated with drug concentrations as indicated in 100 µL culture medium and incubated for 24 h at 37 °C, 5% CO_2_ and humidified air (95%). The assay was performed as recommended by the vendor, and relative luminescence units were recorded using the Fluostar Omega plate reader (BMG).

### 4.7. Neurite Outgrowth Measurement

To quantitatively assess the neurite outgrowth after patulin-treatment of enteric neurons, a cellular transwell culturing system modified from [57] was used. Cell culture transwell inserts with a 3 micron pore size in 24-well formats (Nunc™ cell culture inserts in multi-well plate) were coated with extracellular matrix gel from Engelbreth-Holm-Swarm murine sarcoma (Sigma Aldrich, diluted 1:5 with culture medium). Drug concentration was set to 0.5 µM in 500 µL culture medium (100 µL cell suspension in insert and 400 µL culture medium in basal chamber). After an incubation period of 24 h, neuronal cell bodies from the upper surface of the insert were removed with a cotton swab. Neurites were subsequently stained with crystal violet solution (Sigma Aldrich). For extraction of the dye, inserts were transferred into 400 μL extraction buffer and shaken for 5 min at 600 rpm at room temperature. To quantify neurite outgrowth, 100 µL of extracted stain solution was measured at OD595 and normalized to OD340 (Biochrom ASYS Hitech Expert 96 UV Microplate Reader).

### 4.8. Growth and Viability Testing of Exemplary Murine Fecal Bacteria

The potential impact of the selected extract on bacterial growth was investigated by using freshly collected fecal pellets of wild type mice (C57B6/J OlaHsd, Envigo, 50% male). Samples were diluted with isotonic sodium chloride (0.9%, 100 μL/mg) and homogenized. Fecal suspensions were incubated with either the extract (5 ng/μL), patulin (22 µM) or DMSO as solvent control for 10 min at room temperature with mixing every 2 min. After additional appropriate dilution of the samples, they were spread on 3M^TM^-Petrifilm^TM^ plates specific for Enterobacteriaceae or Lactobacteriaceae (3M Deutschland GmbH, Neuss, DE), 1 mL each. Finally, the plates were incubated for 20 h at 37 °C, colony forming units counted, and normalized to values obtained for DMSO-control. Finally, 1:100 dilutions of the homogenate were additionally used in technical duplicates for viability measurement using the BacTiter Glo assay (Promega, Mannheim, Germany).

### 4.9. Identification of P. coprobium, Fractionation of Its Extract and Verification of Patulin

The producer strain IBWF D03003 was identified as *Penicillium* by morphology. The determination of the species was done by ITS sequencing. The primers ITS 1 and 4 were used for PCR and sequencing. The ITS sequence of IBWF D03003 has 100% identity with three CBS strains of *Penicillium coprobium* (CBS 129797; CBS 561.90; CBS 210.70). The fungal extract was analyzed and fractionated as described previously [20]. Briefly, 400 µg of extract were injected and fractions separated on HPLC to obtain 92 fractions comprising eluent of 15 s. Subsequently, fractions were evaporated and stored at −20 °C until further use. Fractions with activity were reanalyzed using HPLC/MS (Series 1260 Infinity, Agilent, Waldbronn, Germany) equipped with DAD and mass spectrometer (Agilent Qadrupole). A Superspher 100 RP-18 column (125 × 2mm; 4 µm, Merck KGaA, Darmstadt, Germany) was used at 40 °C and a flow rate of 0.45 mL/min. Elution was performed with a gradient of H_2_O and acetonitrile. UV-vis and mass spectra, as well as retention time, were used to compare compounds. Patulin was purchased from Sigma-Aldrich as standard. To determine the amount of patulin in the culture fluid extract of IBWF D03003, a calibration was done with the standard. Sixty-eight percent of the total amount of extract was patulin.

## Figures and Tables

**Figure 1 molecules-24-02776-f001:**
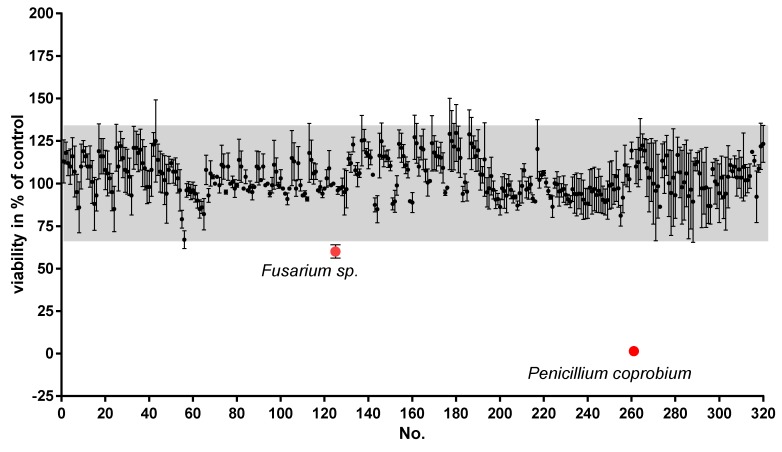
Effects of fungal extracts on viability of human neuroblastoma cell line SH-SY5Y.

**Figure 2 molecules-24-02776-f002:**
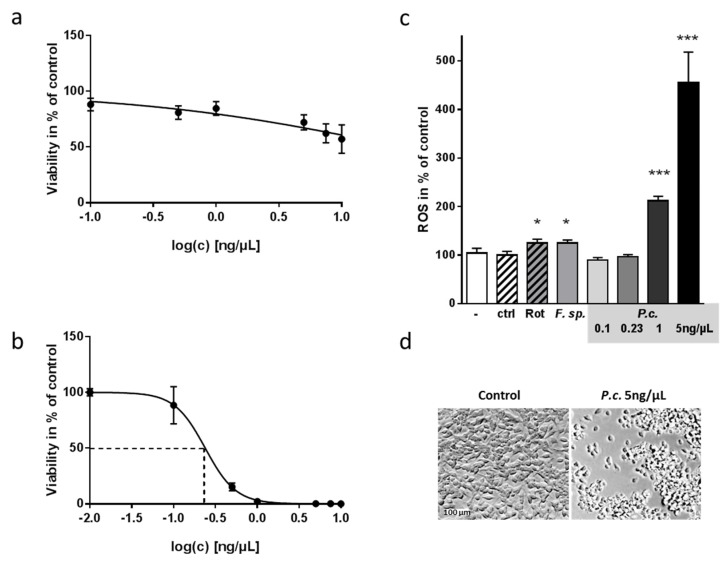
Dose-response curves of toxic fungal extracts from *Fusarium sp.* and *P. coprobium* and effect on production of reactive oxygen species in SH-SY5Y cells. (**a**,**b**) Cells were incubated with the extracts at concentrations as indicated for 24 h (a: *Fusarium sp.*; b: *P. coprobium* extract). Viability was assessed within two independent experiments (*n* ≥ 5). Shown are means ±SD in % of the cells treated with DMSO (1%; solvent). (**c**) To examine the underlying mechanism of toxicity, a luciferase-based ROS production assay was performed. Rotenone served as a positive control (5 nM) and DMSO (1%) as a solvent control (ctrl). *Fusarium sp.* extract was used at 5 ng/µL; *P. coprobium* extract at the indicated concentrations. Data are derived from two independent experiments (*n* ≥ 5); values are represented as mean + SD. Statistical analysis was performed with One Way Anova (* *p* < 0.05, *** *p* < 0.001). (**d**) An exemplary miscroscopic picture of cells treated with solvent or with 5 ng/µL *P. coprobium* is shown.

**Figure 3 molecules-24-02776-f003:**
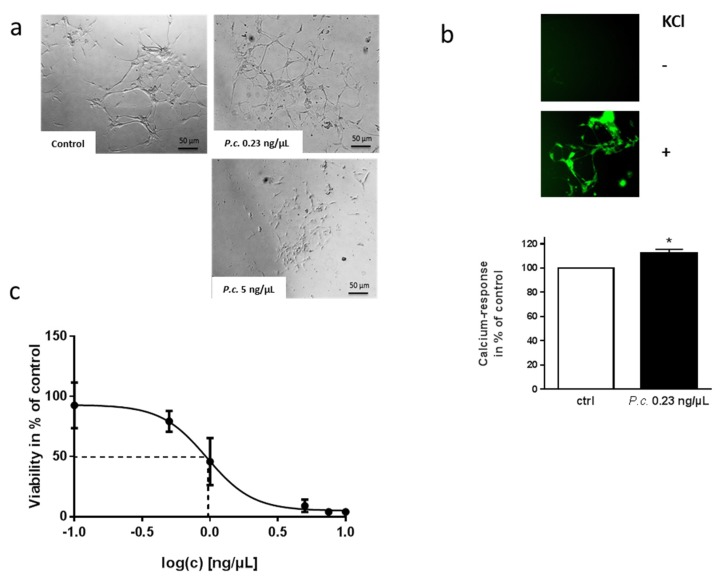
Effect of *P. coprobium* extract on murine enteric neurons and members of the gut microbiota. Murine primary neurons (DIV4) were incubated with the extract at concentrations as indicated for 24 h. (**a**) Cells were investigated by microscopy. (**b**) Cells were loaded with a calcium-sensitive stain and then depolarized with KCl (50 mM). Increase of fluorescence intensity of the stain due to calcium influx (exemplary microscopic picture) was detected by orbital measurements and values normalized by baseline (before stimulation) were set in relation to control (ctrl, 1% DMSO-treated cells, 100%). Indicated are means +SEM (*n* ≥ 4). (**c**) Viability was assessed with the lytic assay using the indicated concentrations (*n* = 4). Values are presented as means +SEM in % of the cells treated with DMSO (solvent). Statistical analysis was performed with One Way Anova (* *p* < 0.05).

**Figure 4 molecules-24-02776-f004:**
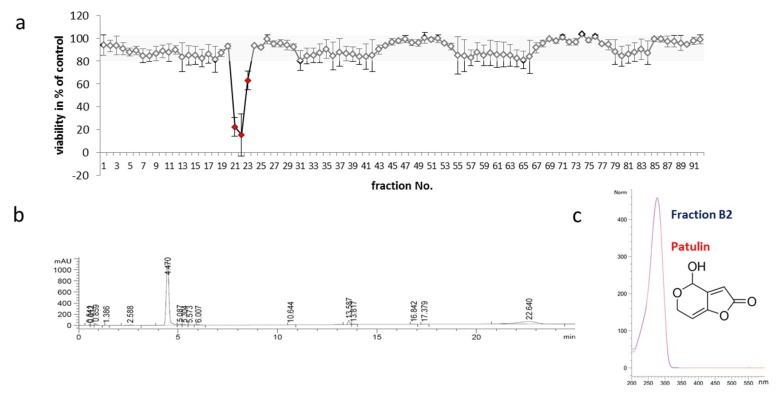
Identification of patulin as one of the effective compounds. The extract from *P. coprobium* was fractionated by HPLC (400 µg in 84 fractions, each eluted within 15 s). (**a**) Fractions were dissolved in DMSO and diluted in DMSO as before and SH-SY5Y cells treated with the fractions or with solvent for 24 h. Viability was measured as described before from two independent fractionations in two repetitions (*n* = 4) with 4 controls (1% DMSO) on each plate. Values are presented as mean ±SD; values obtained for fractions B2-B4 are indicated in red. (**b**) Fractions of interest (B2–B4) were re-analyzed by HPLC/MS (an example of fraction B2 is given, chromatogram at 210 nm) and (**c**) spectra compared to library-deposited spectra (comparison of fraction B2 to patulin; chemical structure of patulin is given).

**Figure 5 molecules-24-02776-f005:**
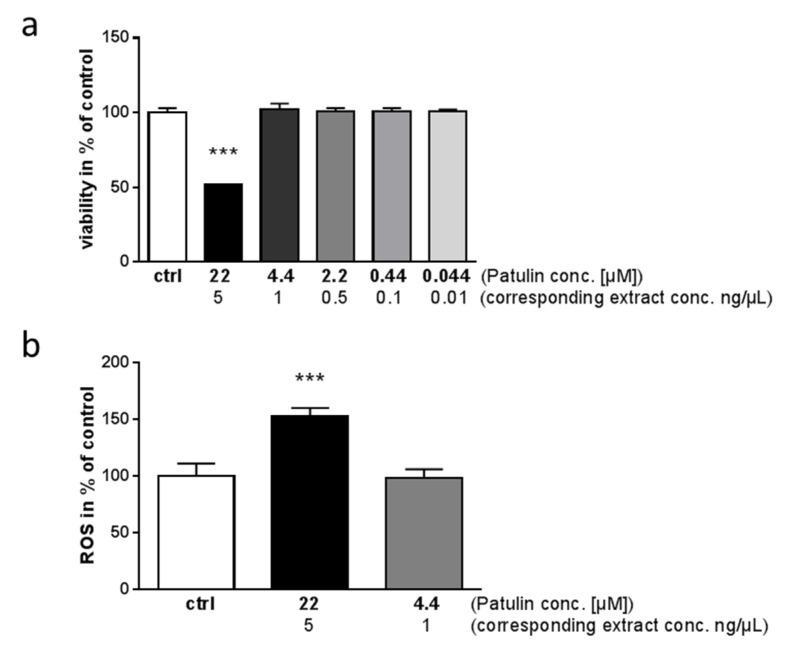
Patulin induces toxic effects and ROS production in SH-SY5Y cells. (**a**) Toxicity of patulin in SH-SY5Y cells was determined as described before for the extract (One Way Anova; *** *p* < 0.001; *n* = 4). Tested concentrations and corresponding extract concentrations are indicated. (**b**) Production of ROS was assessed with patulin at two different concentrations in SH-SY5Y cells (One Way Anova; *** *p* < 0.001; *n* = 4).

**Figure 6 molecules-24-02776-f006:**
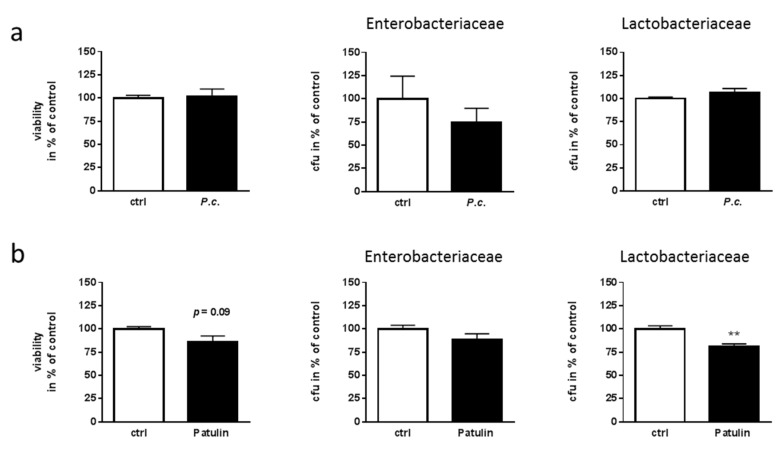
*P. coprobium* extract and patulin reveal no major effects on fecal microbiota. To assess a potential effect on fecal microbiota, freshly obtained feces from wild type donor mice was homogenized and treated with the extract (**a**; 5 ng/µL) or patulin (**b**; 22 µM) before assessing viability (left graph) or before plating on Enterobacteriaceae- and Lactobacteriaceae-selective plates. Colony forming units (CFU) were counted after 20 h of incubation at 37 °C (*n* = 4 for *P.c.*; *n* = 6 for patulin; mean +SEM). Statistical analysis was performed with Student’s t-test.

**Figure 7 molecules-24-02776-f007:**
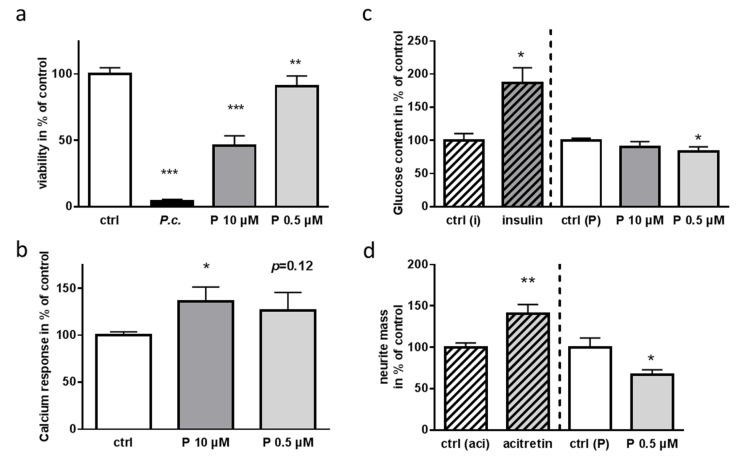
Impact of patulin on functional pathways of primary enteric neurons of the mouse. Primary murine neurons (DIV4) were incubated with the extract (containing 22 µM patulin) or purified patulin (P) at concentrations as indicated for 24 h (*n* ≥ 6). DMSO served as solvent control (ctrl). (**a**) Viability was measured as described before, values represent mean ±SD. (**b**) Calcium response was assessed as described in Figure 3. (**c**) Glucose content in cells was measured by a luminescent assay and normalized to solvent-treated cells (ctrl (P)). As a positive control, cells were treated for 1 h with 1 nM insulin or respective control (ctrl (i)). Statistical analysis was performed with One Way Anova (* *p* < 0.05; ** *p* < 0.01, *** *p* < 0.001). (**d**) Neurite outgrowth was measured by crystal violet stain of transwell-inserts used for enteric neuron cultivation. As a positive control for measuring neurite mass, the synthetic retinoid acitretin (1 µM) or the respective control (ctrl (aci)) were applied for the last 2 days of culture. Statistics were performed by using Student’s t-test for pairwise comparison (* *p* < 0.05; ** *p* < 0.01). All experiments were conducted with at least 4 donor animals.

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
