# Peer review of "Identification of Patulin from Penicillium coprobium as a Toxin for Enteric Neurons"

_molecules, 2019, doi:10.3390/molecules24152776_

Round 1

Reviewer 1 Report

This resubmitted manuscript has been fully considered the prior concerns, and to my opinion, have improved the rigor and quality of the study.

 Minor suggestions,

1) Since patulin increases ROS in the neurons that is a significant impact. This should be stated in the abstract.

 2) Can the authors discuss more on the negative results on fecal bacteria? How does this link to the gut-brain-axis connection? It seems that the style of writing in current version only focuses on neuronal functions.

Author Response

Since patulin increases ROS in the neurons that is a significant impact. This should be stated in the abstract.

Thank you for your kind comment. We agree that due to all the recommendations and valuable suggestions, our manuscript definitely improved in quality and are thankful for your work. In regard to your comments, we now added this information in the abstract as follows:

“Further spectral analysis revealed that the effective compound was patulin and that this polyketide lactone is not only capable of evoking ROS production in SH-SY5Y cells but also diverse functional disabilities in primary enteric neurons such as altered calcium signaling.” (line 27)

Can the authors discuss more on the negative results on fecal bacteria? How does this link to the gut-brain-axis connection? It seems that the style of writing in current version only focuses on neuronal functions.

You are right that there was only minor information and discussion about a potential impact on gut microbiota. We restricted this to a minimum as the extract itself had no impact on ATP level of fecal bacteria and growth of selected genera. We now included also investigation of purified patulin on bacteria due to the recommendation of reviewer 2 (Figure 6b) and also inserted a short part in the discussion regarding this item (line 295ff).We did not give further information about potential impact on gut-brain-axis as this would have gone beyond the scope as we mainly focused on gut neurons. However, the impact of patulin especially on Lactobacteriaceae is absolutely of interest in this regard as these bacteria have been correlated e.g. with depression or anxiety.

Reviewer 2 Report

I'm considering that the content of manuscript was improved compared to previous form. However, it seems that the substance of manuscript became kind of messy with the correspondence against suggestions from reviewers without the reconstruction of context. I request authors to arrange and unify the content including experimental data through whole manuscript.

Comments:

1. If authors would like to keep Fig.3d, same experiment should be done by using patulin because it is considerable that patulin may have the effect to bacteria/micobacteria and some components containing the extract from P. coprobium may mask the bioactivity of patulin.

2. It seems that the data which showed the cytotoxic and ROS productive effect of extract from Fusarium sp.in Fig.1 is not necessary for this study because authors almost did not mention about it in the manuscript. If authors would like to keep this data, some experiments using the extract from Fusarium sp. should be done because that extract demonstrated significant effect on ROS production on SH-SY5Y at least. Otherwise, authors need to show the reasonable explanation why Fusarium sp. was excluded as a research object in this study. Please reconsider how to handle this data.

3. In Fig. 3b, please show the fluorescence image when the extract from P. coprobium was treated to primary cultured ENS.

3. In Fig. 3C, it should be deleted both of gray arrows because it is no meaning to indicate the EC50 of extract from P. coprobium on SH-SY5Y in dose-dependent curve in primary cultured ENS. It is enough to mention in the text such as results or discussions section.

4. I have no idea why Fig. 4d-e combined with Fig.4a-c. Fig.4a-c shows the results related to extracted furcation. In contrast Fig. 4d-e shows the results of bioactive effect of patulin. It may be better to separate those.

5. The effect of the extract from P. coprobium to neurite mass and glucose content on primary cultured ENS have not been examined. I recommend authors to demonstrate those experiments by using the extract from P. coprobium. Otherwise, it is difficult to compare both the extract and patulin bioactive properties.

Author Response

 We are grateful that this reviewer sees the improvement our work gained by considering all the valuable recommendations. We are astonished about his remark about the manuscript becoming “messy” as both other reviewers were absolutely fine with it in its current form despite some real minor changes. We checked the text carefully but cannot agree with your finding despite the new arrangement of Figure 4 which we now divided in Figure 4 and 5. Additionally, we separated the results for bacteria (new Figure 6).This clearly structures the results in a better way and we hope that this also improves structure to your opinion.

Comments:

1. If authors would like to keep Fig.3d, same experiment should be done by using patulin because it is considerable that patulin may have the effect to bacteria/micobacteria and some components containing the extract from P. coprobium may mask the bioactivity of patulin.

 We added investigation about the effect of purified patulin on fecal bacteria within a new sub-figure (Fig 6b, line 233ff and 295ff). As you already considered, some components of P.c. extract seem to have masked a slight but significant toxic effect of patulin on Lactobacteriaceae.

2. It seems that the data which showed the cytotoxic and ROS productive effect of extract from Fusarium sp.in Fig.1 is not necessary for this study because authors almost did not mention about it in the manuscript. If authors would like to keep this data, some experiments using the extract from Fusarium sp. should be done because that extract demonstrated significant effect on ROS production on SH-SY5Y at least. Otherwise, authors need to show the reasonable explanation why Fusarium sp. was excluded as a research object in this study. Please reconsider how to handle this data.

 It is rather common to start from a screening with few candidates and some initial experiments with all of them (e.g. ROS measurement) but later on, focus just on the most promising (in our case P. coprobium extract). We agree with you that our decision did not get clear from our manuscript and therefore now inserted a comment in line 120ff: “As the effect exerted by Fusarium sp. was comparably low and P. coprobium extract seemed to be a more promising candidate for impacting neuronal cells, we focused in the following experiments only on the latter.”

3. In Fig. 3b, please show the fluorescence image when the extract from P. coprobium was treated to primary cultured ENS.

We understand that you would like to have visualization of the obtained results. However, calcium influx is a rather short signaling event. Full reaction in a culture (raise from baseline to maximal value) mostly happens within 9-10 sec, afterwards bleaching occurs if excitation is kept active. Therefore, capturing differences in calcium influx via microscopic pictures is merely not possible using a manual application of KCl. When measuring fluorescence for gaining our results, we use an automated injector and a fluorimeter which is suited for time-lapse experiments. This then gives no microscopic pictures but fast orbital averaging of emission due to the whole well.

3. In Fig. 3C, it should be deleted both of gray arrows because it is no meaning to indicate the EC50 of extract from P. coprobium on SH-SY5Y in dose-dependent curve in primary cultured ENS. It is enough to mention in the text such as results or discussions section.

 We aimed at helping the reader to better understand the differences obtained for both cell lines. However, due to your suggestion we removed the arrow heads in the figure and the respective comment in the figure legend (line 155 ff). We extended our remark in the results description slightly (line 175).

4. I have no idea why Fig. 4d-e combined with Fig.4a-c. Fig.4a-c shows the results related to extracted furcation. In contrast Fig. 4d-e shows the results of bioactive effect of patulin. It may be better to separate those.

We separated Figure 4d-e now from the fractionation (new figure 5).

5. The effect of the extract from P. coprobium to neurite mass and glucose content on primary cultured ENS have not been examined. I recommend authors to demonstrate those experiments by using the extract from P. coprobium. Otherwise, it is difficult to compare both the extract and patulin bioactive properties.

You are right that such a direct comparison is missing. However, within 10 days of revision time, it is not feasible to perform the recommended experiments. In addition, this would need additional at least 4 mice to be sacrificed. Therefore, we considered not to perform these new experiments.

Reviewer 3 Report

The revised manuscript showed appropriate responses to the reviewer comments which improved the quality of manuscript. Therefore, manuscript is recommended for publication.

Author Response

Thank you for your kind comment. We agree that due to all the recommendations and valuable suggestions, our manuscript definitely improved in quality and are thankful for your work.

Round 2

Reviewer 2 Report

Authors have explained and answered to my questions and requests properly. Therefore, I am considering that this manuscript is suitable to be published.

This manuscript is a resubmission of an earlier submission. The following is a list of the peer review reports and author responses from that submission.

Round 1

Reviewer 1 Report

The manuscript by Brand et al. described a study aiming to investigate mycobiome and their metabolites relevant to “the gut-brain-connection”. They reported a screen of 320 fungal extracts in SH-SY5Y cells for cytotoxicity and identified that a compound called patulin from Penicillium coprobium is neurotoxic. The authors further showed that patulin is toxic to primary enteric neurons by increasing cellular ROS and calcium influx as well as changing glucose uptake. The research subject is interesting, however some experimental methods and data presented are neither clear nor technical sound to support major conclusions and claims. To my view, the manuscript in current form is not mature for publication. Additional experiments are essentially needed, and the authors should do proper data analysis and interpretation.

Major Technical Problems and Comments:

1)      In section 2.1 and figure 1, the rationale behind the criteria and threshold for selecting hits is unclear. The authors stated that they used “a value <100 %-3xSD % and a SD between experiments of <10 %”, but no reference cited. A published statistical method should be used.

2)      The ROS data in figure 2 are bizarre, where a high-dose extract has less ROS production than a low-dose one. If the toxicity of the extract is an issue, the authors should validate their assay and use a dose range of the extract tolerable in the cells.  

3)      There are no reference compounds as positive controls in cell viability, Ca2+ and glucose assays. These assays should be proper validated.

4)      Since SHSY5Y cells and primary enteric neurons are different. The authors should do a dose-response quantitative assay to determine a suitable dose of the active extract in primary enteric neurons. It is unwise to directly apply the EC50 dose from one cell type to another.

5)      The methods and characterization data for compound identification are not valid. Only based on UV spectra and HPLC retention time, it is not conclusive to confirm the compound that is patulin. The authors mentioned that a library-deposited spectra database was used but no details in the Methods Section. More sophisticated approaches should be used (i.e., NMR and HR-MS). In fact, the data shown in fig 5A indicated that the crude extract has a higher toxicity than the pure patulin, suggesting that there are other more toxic but unidentified compounds rather than patulin responsible for the effect.

6)  Labels in figure 3d, should the authors use P.c extract instead of patulin in the Enterobacteriaceae assay?

Reviewer 2 Report

Strengths:

In this study, authors tried to analyze 320 fungal extracts and their effect of viability of neuronal cells. Among 320 extracts from fungal, authors identified that the extract from p. coprobium decreased the viability on neuronal cells including primary enteric neurons. In addition, authors clarified that the active compound in extract from p. coprobium for decreasing cell viability was patulin. These results might be help to understand the relationship between fungal commensals of human gut and “gut-brain-axis” and establish the preventing strategies for intestinal injury.

Comments:

1. Authors demonstrated the effect of extract from p. coprobium to the viability on SH-SY5Y generated from central nervous neuron and primary enteric neurons. According to the results, it seems that the toxicity of extract from p. coprobium for SH-SY5Y was more intense than that for primary enteric neuron though the title of this manuscript insists that the extract from p. coprobium including patulin has a toxicity for enteric neurons. Please discuss which type of neurons (from CNS or ENS) are more sensitive against the extract from p. coprobium and patulin.

2. In Fig. 2c, it is obvious that both 1 ng/μL and 5 ng/μL extract from p. coprobium treatment almost killed SH-SY5Y. This means that ROS generated in live cells could not measure. What kind of ROS did authors measure?

3. Did authors measure ROS production when cells were treated with patulin? Please show that results.

4. Did authors explore the effect of patulin on SH-SY5Y? Was the profile of patulin same with that of the extract from p. coprobium on SH-SY5Y?

5. In Fig.5d, why did not authors explore the effect of 10 μM patulin for neurite outgrowth on primary enteric neurons?

6. It seems that Fig.3d is not necessary for this manuscript because this data is almost out of focus for this study. Please reconsider about this data.

7. How did authors set the concentration of patulin for all experiments? Did authors measure the concentration of patsulin in the extract from p. coprobium? Authors need to show the reason why these concentrations of patulin were used in experiments.

8. Authors should discuss about the effect of extract from Fusarium sp.

Reviewer 3 Report

Manuscript entitled "Identification of Patulin from P. coprobium as a toxin for enteric neurons" by Brand et al. report systematic approach for identification of a toxic which affect enteric neurons. The identification and characterization of compound is well reported by author which seems very interesting studies to recommend for publication. The gut micro-organism are key individual for their role to affect peripheral neuronal network (as also known as enteric nervous system) which in turn affect human health. SH-SY5Y human neuronal cell lines were used to see the effect of about 320 fungal extracts and one of the extract from fungus P. coprobium showed toxicity to these neuronal cells at concentration (EC50=0.23 ng/uL) which showed a significant finding in the manuscript. The extract was further characterize by HPLC, LC-MC and confirmed from available database to be patulin. Patulin is reported as secondary metabolites from other fungus as well. .Author further used patulin to see effect on different functional pathway using mouse enteric neurons. Some minor comments for author to consider: 1. Put % DMSO concentration used in the assay in the graphs so that reader can understand well. Just DMSO could be read as 100% DMSO which is toxic to cell. 2. Why author used mouse enteric neurons to see the functional pathway? Why not human? Please add justification in the manuscript. 3. Why not author characterized HPLC pure fraction further using 1H/13C NMR spectroscopy to confirm the structure of compound along with MS data to be Patulin.